# Multilayer In Vitro Human Skin Tissue Platforms for Quantitative Burn Injury Investigation

**DOI:** 10.3390/bioengineering10020265

**Published:** 2023-02-17

**Authors:** Sean Brocklehurst, Neda Ghousifam, Kameel Zuniga, Danielle Stolley, Marissa Nichole Rylander

**Affiliations:** 1Biomedical Engineering, University of Texas in Austin, 204 E Dean Keeton St, Austin, TX 78746, USA; 2Mechanical Engineering, University of Texas in Austin, 204 E Dean Keeton St, Austin, TX 78746, USA; 3Department of Interventional Radiology, UT MD Anderson Cancer Center, Houston, TX 78746, USA

**Keywords:** in vitro model, human skin burns, Arrhenius model, high temperature

## Abstract

This study presents a multilayer in vitro human skin platform to quantitatively relate predicted spatial time–temperature history with measured tissue injury response. This information is needed to elucidate high-temperature, short-duration burn injury kinetics and enables determination of relevant input parameters for computational models to facilitate treatment planning. Multilayer in vitro skin platforms were constructed using human dermal keratinocytes and fibroblasts embedded in collagen I hydrogels. After three seconds of contact with a 50–100 °C burn tip, ablation, cell death, apoptosis, and HSP70 expression were spatially measured using immunofluorescence confocal microscopy. Finite element modeling was performed using the measured thermal characteristics of skin platforms to determine the temperature distribution within platforms over time. The process coefficients for the Arrhenius thermal injury model describing tissue ablation and cell death were determined such that the predictions calculated from the time–temperature histories fit the experimental burn results. The activation energy for thermal collagen ablation and cell death was found to be significantly lower for short-duration, high-temperature burns than those found for long-duration, low-temperature burns. Analysis of results suggests that different injury mechanisms dominate at higher temperatures, necessitating burn research in the temperature ranges of interest and demonstrating the practicality of the proposed skin platform for this purpose.

## 1. Introduction

Skin burn injuries often result in significant physical and psychological damage that can be devastating to quality of life and lead to lifelong health risks [1,2,3]. Despite the prevalence and severity of such injuries, the damage caused by a specific burn cannot be accurately predicted without a greater understanding of the underlying mechanisms of injury for specific tissues. The kinetics of burn injuries rely on the underlying thermal injury reaction rates that result in damage that is highly dependent on the burn temperature, with distinct mechanisms of injury occurring at different exposure temperatures. Thus, to provide meaningful data for the evaluation and prediction of burn injuries, both the time–temperature history and tissue damage must be quantified simultaneously for the same experiments and under the same conditions [3,4,5]. However, this becomes more complex when considering burns caused by high temperatures over short time scales due to the large spatial and temporal temperature gradients involved that cannot be readily quantified because of the difficulty of quantifying evolved temperatures. Historically, studies of skin burns have primarily focused on qualitative observations from histology and protein expression, with limited quantitative results for low-temperature, long-duration hyperthermia, excluding the study of high-temperature, short-duration contact burns [4,6,7]. Therefore, there is a significant gap in research involving the quantitative characterization of the effects of wide-ranging temperatures on tissues under realistic conditions [3,8]. Such data would enable the development of comprehensive computational predictive models for skin burn tissue response that could help achieve positive patient outcomes while minimizing complications due to damage to the surrounding tissue [7,9,10,11,12].

To date, evaluations of burn injury from both contact burn and hyperthermia studies have primarily relied on the Arrhenius model to account for the temperature dependence of injury rates, with the common form assuming irreversible first-order kinetics [6,13]. Two process coefficients determine the behavior of the standard form of the Arrhenius model: frequency factor (A, Hz) and activation energy ($E_{a}$, kJ/mol). The cumulative effective minutes of exposure at 43 °C (CEM43), the current gold standard used to compare a wide variety of burn profiles, is derived from the Arrhenius model. This isoeffective dose metric extrapolates time–temperature histories to one value that can account for comparable injury between various burn conditions, regardless of differences in time–temperature history [4,5,7,11,14]. However, these models still simplify complex burn injury processes, as they do not account for the difference in injury mechanisms active in different temperature ranges, which must be verified experimentally [6,15].

Unfortunately, the development of more complex models of skin burns relies on the availability of comprehensive quantitative datasets of burns injury, which has been historically limited due to the underlying difficulties of directly measuring spatial temperature distribution and corresponding spatial tissue response. Many studies have modeled burns computationally using finite element modeling to calculate the distribution of heat in samples [10,16,17]; however, a lack of accompanying experimental data from biological models limits the utility of such models alone in determining burn injury mechanisms and the extent of tissue damage [10,16,17]. 

Moreover, a wide range of biological skin models have been implemented to quantify burn injuries of the skin, including two-dimensional (2D) in vitro cell monolayers, in vivo animal models, human and animal ex vivo tissues, and, most recently, three-dimensional (3D) in vitro skin platforms [18,19,20,21,22,23,24,25,26]. Both in vivo and ex vivo animal models and human ex vivo tissue have been utilized extensively to study the overall process of burn injury and wound healing [26,27,28,29,30,31,32,33,34]. While much knowledge can be gained from animal models, these in vivo and ex vivo studies are expensive, time-consuming, and ethically complicated, and ultimately, the results have limited applicability to humans due to differential sensitivity to high-temperature burns [26,35]. Excised human skin, which is often collected incidentally during other surgeries and maintained ex vivo, provides the most physiologically representative model. However, such models are in limited supply and highly prone to the variability of factors such as tissue viability, growth conditions, and donor health, making them too inconsistent to obtain the quantitative data necessary for fitting Arrhenius parameters [4,22,36]. Two-dimensional in vitro cell monolayers have been used for burn research due to their simplicity and controllability, facilitating the collection of significant and reliable quantitative data. However, 2D models are less physiologically representative than other models and fail to account for cell–extracellular matrix (ECM) interactions and spatial effects, which can have a significant impact on the response of the tissue to burn exposure, limiting the applicability of the quantitative data generated from these simplified models [3,15,37,38,39].

The need for a physiologically representative, high-throughput model for human skin burn studies has led to the recent emergence of engineered 3D in vitro skin platforms. These in vitro models have provided a new avenue for burn research due to their consistency, customizability, and ease of use, as well as the ability to incorporate various human cells in a more relevant tissue microenvironment [4,9,40]. Prior studies have utilized 3D platforms to investigate the burn wound healing process with fibroblasts and keratinocytes, the dominant cell types in skin, embedded within 3D collagen I hydrogels [22,24,25]. These studies have induced injuries through high-temperature, short-duration contact burns to characterize the tissue response over time with histology and immunohistochemical analysis to visualize cross sections of burn injuries and measure re-epithelialization and the expression of proliferation markers, epidermal differentiation markers, and inflammatory mediators [22,41,42]. However, existing 3D skin platform studies have not sufficiently quantified temperature profiles or thermal properties, resulting in a lack of correlation between temperature profiles and the corresponding skin response.

To address this existing gap in research, we developed and characterized a 3D avascular in vitro skin tissue platform consisting of human dermal fibroblasts and keratinocytes embedded into collagen hydrogel layers to mimic the dermis and epidermis, respectively. We characterized cell viability, cell distributions, specific heat capacities, thermal conductivities, and compression moduli of full-thickness multilayer skin tissue platforms and their subcomponents over time. We utilized the skin platform as a burn research model combined with computational modeling to determine the temperature distributions and injury within platforms at much higher temporal and spatial resolutions than could be directly quantified [5,43,44]. Arrhenius rate coefficients were determined to characterize collagen degradation through prolonged hyperthermia at 47 and 50 °C. In addition, contact burns were induced with a copper tip heated to 50, 65, 75, 85, and 100 °C exposed to samples for three seconds to produce injuries with severities ranging from superficial first-degree to full-thickness third-degree burns. The resulting injury was determined by measuring distributions of heat shock protein 70 (Hsp70) expression, total cell death, apoptosis markers, and tissue ablation [7,8,45,46,47]. The Arrhenius model was applied to contact burn testing and temperature modeling results to determine rate coefficients for the different mechanisms of burn injury. The threshold values for CEM43 were determined from the experimental results to correlate with the resulting injury to evaluate the applicability of CEM43 for prediction of high-temperature, short-duration contact burns. The in vitro skin platforms developed in this study demonstrate the potential to be burned in a controlled fashion to yield different physiologically representative degrees of burn injuries with known temperature distributions. The developed in vitro skin models and the complementary concurrent computational modeling provide a platform to investigate a wide range of contact burn and hyperthermia studies, including measurement of changes in cell protein expression and the activation energy of cellular damage. Ultimately, the outcome of this study will help us better understand the involved cascades of injury post burn injury, which would be beneficial in evaluating and determining the effects of heat-based therapies on human skin. 

## 2. Material and Methods

### 2.1. In Vitro Skin Platform Preparation

#### 2.1.1. Cell Culture

Cryopreserved normal human dermal fibroblasts (NHDFs, PromoCell C-12302, PromoCell, Heidelberg, Germany) and immortalized human dermal keratinocytes (CCD 1106 KERTr, ATCC CRL-2309, ATCC, Manassas, VA, USA) were cultured at 37 °C with 5% CO_2_. Fibroblasts were cultured in Fibroblast Basal Medium 2 (PromoCell C-23220, PromoCell, Heidelberg, Germany) with SupplementPack Fibroblast Growth Medium 2 (PromoCell C-39320, PromoCell, Heidelberg, Germany). Keratinocyte serum-free medium (Gibco 17005-042, ThermoFisher Scientific, Grand Island, NY, USA) with added keratinocyte supplements (Gibco 37000-015, ThermoFisher Scientific, Grand Island, NY, USA) was used for keratinocyte culture. Cells were grown to 80–90% confluence to construct the skin platforms.

#### 2.1.2. Preparation of Collagen

Collagen type I was utilized for the skin platform’s extracellular matrix and was sourced from rat-tail tendons using the isolation method outlined by Szot et al., yielding a lyophilized solid [47]. Collagen type I is a primary constituent of human skin, and collagen I hydrogels have been characterized and are commonly utilized in in vitro studies of engineered skin and other tissue types [13,25]. Lyophilized collagen was then dissolved in 0.1% glacial acetic acid solution to double the desired concentration (6 mg/mL), then neutralized to pH 7.4 with 1X DMEM, 10X DMEM (Sigma Aldrich, St. Louis, MO, USA), and 1N NaOH (Fisher Scientific, Pittsburgh, PA, USA) to a final working concentration of 3 mg/mL. 

This concentration of collagen was selected as a compromise between dermal substitutes such as Apligraf (0.66 mg/mL) and the platforms used in previous studies [47,48,49,50], such as 5 mg/mL used by Helary et al. [49]. In several previous studies, researchers have created skin platforms using collagen concentrations of 2-4 mg/mL, with 3 mg/mL often used for this purpose [49,51]. 

#### 2.1.3. Single-Layer Skin Platform Creation

Single-layer platforms were created with either fibroblasts or keratinocytes to mimic the dermis and epidermis layers of skin, respectively. Thermal and mechanical properties and cell viability were characterized independently on both types of single-layer platforms to compare the potential impact each cell line would have on outcomes and verify the consistency of these properties in experiments. Collagen I hydrogels were developed as previously described with cells suspended and mixed uniformly within the neutralizing buffer solution to a final concentration of 1 × 10^5^ cells per mL. The resulting cell–collagen mixture was then pipetted (480 μL per well) into solid 24-well plates and polymerized for 30 min at 37 °C before the addition of each cell type’s respective cell medium (400 μL) on top of the hydrogel [52,53]. Platforms measured 15.6 mm in diameter and 1.5 mm in thickness after accounting for shrinkage. A thickness of 1.5 mm lies within the range of human skin thicknesses found in previous studies (0.35–2.55 mm, 1.55 ± 0.25 mm or 2.56 ± 0.39 mm depending on skin source location and donor gender) [54,55]. Comparable acellular platforms were created to serve as a baseline to quantify changes in compression moduli and thermal properties of cell-free platforms over time. All in vitro skin platforms were maintained at 37 °C with 5% CO_2_, and cell medium was changed every other day.

#### 2.1.4. Multilayer Skin Platform Creation

Multilayer platforms consisting of a keratinocyte-bearing epidermis layer above a fibroblast-bearing dermis layer were created using multiple iterations of the single-layer gel-making process described above. The first layer was cast and allowed to polymerize for 30 min. The next layer was then made in the same fashion and pipetted directly on top of the previous layer. The layers within platforms created in this fashion adhered and did not delaminate, even with rough handling. Fibroblast Basal Medium 2 was used to maintain multilayer platforms. The relative thickness of the dermis and epidermis layers varies significantly depending on the region of the body. The thicknesses used in this study for the dermis and epidermis (1.125 and 0.375 mm, respectively) are based on in vivo ultrasound measurements of the skin on human volunteers’ hands [56].

### 2.2. Platform Characterization

#### 2.2.1. Cell Viability Testing

Cell viability within skin platforms was quantified to confirm the health and growth rate for untreated platforms 1, 3, 5, and 7 days post sample preparation to establish a baseline prior to burn tests using a CellTiter Blue assay (Promega G8081, Promega, Madison, WI, USA) according to the manufacturer’s procedure.

#### 2.2.2. Spatial Distribution of Cells

Immunocytochemistry staining was performed to observe the distribution of each cell type and growth progression in untreated platforms. Skin platforms were rinsed twice with phosphate-buffered saline (PBS, Fisher Scientific BP661-50, Fisher Scientific, Waltham, MA, USA), followed by fixation with 4% paraformaldehyde solution for 15 min and rinsing with 1X PBS. Platforms were then permeabilized with 0.1% Triton-X 100 (Sigma Life Science X1001-1L, Sigma-Aldrich, St. Louis, MO, USA) for 5 min to stain keratinocytes (not necessary for fibroblasts). Platforms were incubated with a blocking buffer of 5% bovine serum albumin (BSA, Fisher BioReagents BP9700-100, Fisher Scientific, Waltham, MA, USA) in PBS for 1.5 h at room temperature to minimize non-specific bindings. The distribution of fibroblasts and keratinocytes within the platforms was detected using fluorescent anti-human CD90 conjugated with Brilliant Violet 421 (BioLegend 328122, BioLegend, San Diego, CA, USA, 1:20) and anti-human cytokeratin 14 conjugated with Alexa Fluor^®^ 488 (Abcam ab192055, abcam, Waltham, MA, USA, 1:100), respectively. Samples were incubated with antibodies diluted in the blocking buffer for 1 h at room temperature in darkness. Platforms were then rinsed with PBS and imaged with a Leica TCS SP8 confocal microscope using a 10× dry objective (10×/0.6 NA, Leica) at a resolution of 512 × 512 pixels and a z-step size of 4.4 μm.

#### 2.2.3. Mechanical Testing

The compression moduli of in vitro skin platforms (single-layer, multilayer, and acellular) were determined through uniaxial compression conducted 1, 7, 14, 21, and 28 days post sample preparation to determine changes in the stiffness of skin platform extracellular matrices over time, verify consistency between independent experiments, and evaluate matrix degradation from heat exposure. Measurements were conducted with an Instron Electropuls E1000 uniaxial machine. Briefly, each platform was gently removed from its container and placed on a wide, flat stainless-steel surface. The center two-thirds of each platform was isolated with a 9.525 mm diameter leather punch, removing the gel meniscus. A compression platen with a larger diameter than the platforms was pressed down onto platforms at a rate of 0.1 mm/s with position and force collected at 1000 Hz. Mechanical results were validated by testing a series of materials with known elasticity (linear and non-linear).

The distance traveled by the compression plate was divided by the gel height to obtain the strain, and the load was divided by the gel area to obtain the stress. Before testing, the height was found to be 1.5 mm by placing a black non-water-soluble dye on top of the gels and taking images of the profile with a marked scale. The area determined by the diameter of the punch was 71.36 mm^2^. Stress (σ) was determined by dividing the force by the gel area, and strain (ε) was defined as the ratio of the change in height to the original height. True stress (σ’) and true strain (ε’) were derived by assuming a constant gel volume during compression and are related to actual stress and strain by the following relations: σ’ = σ(1 + ε) and ε’ = ln(1 + ε). The compression modulus was determined to be the average slope of the σ’ versus ε’ curve in the range of 5 to 15% ε’.

#### 2.2.4. Specific Heat Capacity and Thermal Conductivity Characterization

Thermal conductivity and specific heat capacity of in vitro skin platforms were determined to confirm the similarity of the parameters to human skin and serve as inputs to the computational model to calculate the propagation of heat through platforms during burn tests. Measurements were taken 1, 3, 5, 7, and 14 days post sample preparation. Differential scanning calorimetry (DSC) was performed on acellular and single-cell platforms to determine the specific heat capacity and the denaturation temperature of platforms. Portions of platforms were weighed out between 5 and 10 mg and placed into separate Tzero pans (TA Instruments, Newcastle, DE, USA), closed with Tzero hermetic lids (TA Instruments, Newcastle, DE, USA), and placed on the reference sensor of the DSC (TA Instruments DSC 2000, Newcastle, DE, USA). Samples were heated from 30 to 80 °C at a rate of 5 °C/min, and liquid nitrogen was used for cooling. DSC results relate to the solid component of skin platforms, as the samples must be dehydrated before testing. The specific heat capacity of hydrated skin platforms was calculated from a weighted average of the specific heat capacities of the liquid and solid components based on density [57]. 

The thermal conductivity of in vitro skin platforms was measured using a custom thermal conductivity measurement device, as shown in Figure 1. Platforms were placed between the two large copper cylinders of the device, with one cylinder heated with a known power and the cold cylinder protruding into an ice bath. A pair of K-type thermocouples recorded the temperature of each element close to the gel being tested, and the temperature was read using a Digi-Sense 20250-01 thermocouple reader. The elements and platforms were housed within a vacuum chamber and held in place using insulative materials to minimize heat loss. A constant power of 0.66 W was input to the system until each element’s temperature reached equilibrium. The material’s thermal conductivity was then calculated from the difference in temperature between the hot and cold elements, the resultant heat flux, and dimensions of the platform using the one-dimensional form of Fourier’s law: $q_{x}=-k\frac{dT}{dx}\;$, where *q_x_* is heat flux (W/m^2^), *k* is thermal conductivity (W/m^2^K), *T* is the temperature in Kelvin, and *x* is the distance from the surface of the hot element (m). Errors in heat flux due to radiative effects and errors in the temperature change due to thermocouple placement were evaluated using an equilibrium finite element analysis (FEA) that modeled our experimental setup in SolidWorks Simulation package (2019). This analysis determined that these potential error sources were insignificant and that the heating element’s power was effectively transferred through the sample.

#### 2.2.5. Characterizing Thermal Collagen Degradation

In order to characterize the collagen matrix breakdown in response to prolonged hyperthermia and determine thermal collagen degradation, platforms were exposed to 47 and 50 °C for up to 30 min. Under these conditions, temperatures can be measured directly with a thermocouple. A needle-point K-type thermocouple with a Digi-Sense 20250-44 temperature data logger was used to measure the temperature of skin platforms during experiments, whereas the temperature of platforms during contact/local burn testing could not be measured with sufficient spatial and temporal resolution. Reductions in compression modulus, measured as described above, were also determined as an indicator of collagen thermal degradation [12] because of the relative simplicity of the degradation mechanism compared to cell death and expression [13,58]. Due to the slow rate of heating and the minimal thickness of the samples, the temperature throughout the platforms can be assumed to be uniform, as confirmed by calculating the Biot number (Bi) of the platforms at less than 0.01. Temperature-versus-time results obtained from prolonged hyperthermia were used to determine the process coefficients of an Arrhenius injury model for thermal collagen degradation using Equation (1):(1)$$\Omega \left(\tau \right)=ln\left(\frac{C\left(0\right)}{C\left(\tau \right)}\right)=\int_{0}^{\tau }A\;e^{\;\left(\frac{-Ea}{R\;T\left(t\right)}\right)}dt$$
where *C*(0)/*C*(τ) is the ratio of the initial compression modulus over the modulus at time τ (s), the logarithm of which represents the damage parameter (Ω) (dimensionless); A is the frequency factor (Hz); $E_{a}$ is the activation energy (kJ/mol); R is the gas constant; T is the temperature (K); and t is time (s) [36]. A damage parameter value of Ω ≥ 1 was defined for the ablated collagen, which is the standard threshold used in numerous previous studies [14,20,59,60]. The process coefficients (A and $E_{a}$) must be determined experimentally for each component of burn injury being evaluated and, ultimately, must be tested for the temperature ranges of interest to validate the relationship. These coefficients were later used as inputs to predict contact burn injury from the computationally modeled time–temperature histories and to evaluate the applicability of the relationship at low temperatures for high-temperature, short-duration contact burns.

### 2.3. Burn Testing

#### 2.3.1. Contact Burn Testing

Platforms were subjected to contact burn injuries by adapting the methodology used by Coolen et al. to create burn injuries with predictable temperatures to characterize both extracellular matrix injury and the response of cells induced by short-duration, high temperature injuries [27]. Briefly, a contact burn apparatus was made with a flat-bottomed 3 mm diameter cylindrical copper element heated to a known controlled temperature using a modified soldering iron and a thermocouple. The burning tip’s vertical position was controlled using a motor and a belt drive, with a microswitch used to set the bottom-most position. An Arduino Uno microcontroller was used to operate the device, allowing for precise control of burn tip temperature, position, and timing for consistent application of heat and consistent resulting burns on the skin platforms. Burn tests were performed with the burn tip heated to 50, 65, 75, 85, or 100 °C for 3 s of contact with the platforms. 

#### 2.3.2. Cell Viability and Apoptosis in Burn Injury

Live–dead calcein/propidium iodide staining was performed to determine the spatial distribution of viable cell within platforms after the burn injury. Twenty-four hours post contact burn, platforms were treated with 1 μM calcein (Thermo Scientific™ C1430, Fisher Scientific, Waltham, MA, USA) and 30 μg/mL propidium iodide (PI, Sigma Aldrich P4864, Sigma Aldrich, St. Louis, MO, USA) solutions for 40 and 20 min, respectively, followed by three rinses with PBS for two minutes per rinse; then, samples were imaged by confocal microscopy rinsing. 

The spatial distribution of apoptotic cells was determined using Annexin V staining method 24 h post burn. Platforms were rinsed twice with 1X Annexin V binding buffer (Biotium 99902, Biotium, Fremont, CA, USA), followed by 30 min incubation with 0.5 μg/mL Annexin V CF488A conjugate (Biotium 29005, Biotium, Fremon, CA, USA) at room temperature. Samples were then rinsed twice with the binding buffer and imaged with confocal microscopy.

#### 2.3.3. Heat Shock Protein 70 Expression in Burn Injury

Heat shock protein 70 (hsp70) upregulation is strongly associated with injury caused by hyperthermia and has been used as an indicator of burn injury in the literature [8,47]. To determine the expression of Hsp70 within platforms in response to contact burns, platforms were fixed with 4% paraformaldehyde 24 h after burning and stained with primary mouse anti-human hsp70 (Abcam ab5442, abcam, Waltham, MA, USA, 1:500) and secondary AlexaFluor 488 conjugated goat anti-mouse (Invitrogen A11001, Fisher Scientific, Waltham, MA, USA, 1:2000). Samples were rinsed with PBS and imaged using confocal microscopy. 

#### 2.3.4. Burn Injury Evaluation

The severity of burn injury to skin platforms caused by contact burns of different temperatures was quantified using volumes of the region: (1) ablated completely, (2) cell death as measured by calcein/PI staining, and (3) apoptotic cells measured by Annexin V staining. The ablated region was visible in all images, regardless of the fluorescent label used. Experimental results were obtained by analysis of confocal microscopy images of skin platforms after contact burning. Images were arranged into stacks of 88 μm thickness, and the areas of the affected regions were measured for each stack using ImageJ (v1.52). Then, volume was calculated using MATLAB (R2018A) from known image positions, stack thicknesses, and areas by approximating the system as a series of concentric truncated cones.

### 2.4. Computational Modeling of Temperature Profile

#### 2.4.1. Calculating Temperature Distributions

Temperature distributions within platforms during contact burns were computationally modeled in Comsol Multiphysics (5.4a) to obtain a much higher temporal and spatial resolution than could be directly measured. The modeled time–temperature histories were correlated with the experimentally determined distributions of injured tissue volume to determine the level of injury caused by each temperature. Modeling was performed using a time-dependent heat transfer study with the experimentally determined thermal characteristics of our platforms. The porous media study preset was not used because the measured value for thermal conductivity was already biphasic.

For the Comsol simulation, the platform’s properties and the tip were set to water and copper, respectively. The tip was inserted 0.2 mm into the gel, and perfect contact was assumed between the gel and the tip. The tip’s initial temperature was set to 50, 65, 75, 85, and 100 °C, matching the contact burn protocol, and the initial temperature of the room and gel was 23 °C. A tetrahedral mesh of minimum element size of 0.075 mm, maximum mesh size of 0.15 mm, maximum growth rate of 1.3, curvature factor of 0.2, and resolution of the narrow regions of 1.0 was chosen after performing our preliminary mesh refinement study. The model was run repeatedly with an increasingly fine mesh until the results converged and did not change with any additional details, ensuring that the results were independent of the meshing parameters. On the side of the tip distal to the gel, an isothermal boundary condition was imposed to represent the large, approximately isothermal metallic body of the shank of the burn apparatus and to optimize the model for runtime. The isothermal surface replenishes the heat lost to the platform, and a temperature gradient is formed across the tip. The inbuilt physics-controlled timestep was used to ensure convergence throughout the simulation, and data were saved every fifty milliseconds for postprocessing and data analysis. The simulation was run for two separate steps; the first step modeled three seconds of contact with the burn tip, and the second step modeled the propagation of residual heat throughout the platforms for one minute after the removal of the burn tip. 

#### 2.4.2. Isoeffective Thermal Dose Calculations

The most commonly used thermal isoeffective dose metric, cumulative effective minutes of exposure at 43 °C (CEM43), was applied to compare time–temperature histories and evaluate this metric’s suitability for high-temperature, short-duration contact burns [8]. A CEM43 value of “X” is considered to be the equivalent of “X” minutes of exposure at 43 °C and was calculated using the following Equation (2):(2)$$CEM43=\;\int_{0}^{t}\;R^{\;\left(43-T\right)}dt\;\;\left(\min\right)$$
where *T* is temperature (°C), *t* is time (min), and *R* = 0.5 for *T* > 43 °C and *R* = 0.25 for *T* < 43 °C. This metric provides a means of estimating the total effective thermal dose received by skin platforms given temperature profiles that change during exposure [4,5,7,11,14]. The spatial distribution of total CEM43 within skin platforms exposed to contact burns was calculated for each burn profile using the temperature distributions calculated in Comsol and the integral Equation (2) presented above. CEM43 was totaled using the full time–temperature history for that particular voxel at every voxel in the simulation. CEM43 was also calculated for the platforms exposed to low-temperature, long-duration hyperthermia using their measured internal temperatures in order to correlate it back to the collagen degradation.

#### 2.4.3. Determining Arrhenius Coefficients for Contact Burns 

The Arrhenius coefficients (A and $E_{a}$) were determined for thermal collagen ablation and cell death at high temperatures using the volume of injured region data from contact burn testing and computational modeling of temperatures. The damage parameter (Ω) was defined as Ω ≥ 1 for collagen ablation and Ω ≥ 2.303 for cell death based on the Arrhenius model Equation (1), where 90% of cells are dead.

For a given set of coefficients (A and $E_{a}$), Ω was calculated for every tetrahedral voxel of the computational model of temperatures over the full time–temperature history of each contact burn protocol. Values for coefficients A and $E_{a}$ were determined such that the total volume of all voxels exceeding the threshold for injury (Ω), as determined using the temperature model, equaled the volumes of injury observed experimentally with contact burns. These values were calculated for all five contact burn tip temperatures and the final values for A and $E_{a}$ by minimizing the sum of the squares of the differences between experimental observations and calculated predictions.

#### 2.4.4. Contact Burn Injury Thresholds and Predictions

Published values for Arrhenius coefficients (A and $E_{a}$) were used to predict injury to skin platforms based on temperature modeling, and these predictions were compared with experimentally observed results of contact burn testing. Moreover, the Arrhenius injury model was calculated with coefficients A and $E_{a}$ (for thermal collagen degradation and cell death, respectively) obtained in the present study and compared with published data [13,21,22,61]. The damage parameter (Ω) was calculated for every voxel within the finite element model for the full time–temperature histories of five different tip temperatures used for contact burn testing. The total volumes with Ω values greater than or equal to the thresholds given by the source of the Arrhenius coefficients were calculated, representing the predicted volume of injured tissue. Conversely, Ω threshold values were found such that the predicted volumes of injured tissue matched the volumes experimentally determined in the present study. This process was also used to find injury thresholds for CEM43 using the standard form of the CEM43 integral described in Section 2.4.2 instead of the Arrhenius model. 

## 3. Results

### 3.1. Characterization

#### 3.1.1. Viability

Figure 2 demonstrates that the cell viability of both cell types, as measured by CellTiter Blue assay, increased over time, as expected, indicating cell survival and growth in the hydrogels. Fibroblasts proliferated slowly and steadily throughout the growth time, increasing by a total of 1.9-fold (*p* < 0.05) from day one to day seven. Keratinocytes grew reasonably slowly at first before accelerating after day 5, increasing by a total of 2.5-fold (*p* < 0.05) from day one to day seven. 

#### 3.1.2. Spatial Distribution of Cells

Figure 3 shows that both fibroblasts and keratinocytes were evenly distributed throughout their respective layers within the skin platforms. They proliferated throughout the seven-day growth period. The morphology of fibroblasts became elongated over time, whereas keratinocyte morphology remained consistently round. Figure 4 shows a profile view of a multilayer skin platform on day seven post sample preparation. Fibroblasts and keratinocytes remained within their distinct dermal and epidermal layers, the interface between which is visible in Figure 4.

#### 3.1.3. Mechanical Properties and Consistency Evaluation

As shown in Figure 5, mechanical compression testing results demonstrated that the compression modulus did not significantly change over time for any in vitro skin platforms. We found no significant difference in compression moduli between the single-layer epidermis containing keratinocytes, the single-layer dermis containing fibroblasts, and the multilayer platforms with both epidermis and dermis layers. There was also no significant difference between the non-celullarized samples and the cell-bearing samples, suggesting that cells did not significantly modify the collagen hydrogels throughout the timeframe of our experiments, minimizing such modification as a potential source of error. The average compression modulus for multilayer platforms with epidermal and dermal layers was found to be 20.0 ± 0.87 kPa. 

#### 3.1.4. Thermal Properties

Differential scanning calorimetry measurements indicate that the specific heat of in vitro skin platforms did not change significantly with longer growth times. The specific heat capacity of dry samples was found to be 3.91 ± 0.07 J/g/°C, from which the average effective specific heat capacity of skin platforms was calculated to be 4.18 J/g/°C.

Thermal conductivity values for each platform were obtained with a custom-built measurement device described in Section 2. Measurements were taken with the temperature not exceeding 25 °C, and no breakdown of hydrogels was observed during testing. There was no significant change in thermal conductivity observed for any time point measured for 1, 3, 5, 7, and 14 days post sample preparation (p=0.69), and the average thermal conductivity measured for all multilayer platforms combined was 0.603 ± 0.128 W/mK. 

#### 3.1.5. Characterizing Thermal Collagen Degradation 

The internal temperatures of vitro skin platforms exhibited a steady increase over time after being placed within an incubator with a set temperature of 47 or 50 °C, with consistent temperature curves between trials. These temperature data were also used to calculate the CEM43 dose for each sample, using Equation (2) shown in Section 2.4.2. As shown in Figure 6, the compression moduli of the acellular platforms decreased significantly (*p* < 0.05) with exposure time, with greater degradation caused by 50 °C hyperthermia compared to 47 °C. After 10 min of exposure, the samples exposed to 47 °C were reduced to 89% of the control with a CEM43 of 50.2, and the samples exposed to 50 °C were reduced to 53% of the unheated sample with a CEM43 781.6. After 15 min, the samples exposed to 47 °C were reduced to 76% of the control with a CEM43 of 105.4, whereas the samples exposed to 50 °C had degraded almost entirely, except for insignificant traces that could not be mechanically tested, with a CEM43 of 1618.

Based on the normalized compression moduli and CEM values, the Arrhenius coefficients for thermal collagen degradation were found to be A= 1.61 × 10^45^ s^−1^ and Ea=2.96 × 10^5^ J/mole. 

### 3.2. Burn Testing and Modeling

#### 3.2.1. Contact Burn Testing Results

Exposure of in vitro skin platforms to 3 s contact burns with tip temperatures between 50 and 100 °C resulted in an immediate partial burn beneath and around the contact point. As shown in Figure 7, the volume of the ablated region increased drastically with higher temperatures of the burn tip. Three-dimensional images of the burned and stained platforms clearly show the shape and size of the burn-injured region. No intact cells were present in the ablated region immediately beneath and around the point of contact with the burn tip, although some dead cells marked by propidium iodide remained in the partially degraded matrix. At the edge of this destroyed region, we observed large groups of dead cells injured by the burn. Three seconds of exposure at 50 °C resulted in a very shallow, superficial burn, whereas a tip temperature of 100 °C burned through the full thickness of the skin platforms. With a tip temperature of 75 °C, burns entirely ablated sections of the platform and partially degraded a region of the collagen matrix.

A region of complete tissue ablation centered beneath the burn tip was visible in the confocal microscopy images for each burn tip temperature above 50 °C. Live/dead imaging using calcein and propidium iodide showed a region of substantial cell death at the periphery of the injury site, following the ablated region’s contour labeled with Annexin V, which survived the initial burn exposure but became apoptotic due to injury. Beyond this is a region of cells expressing Hsp70 due to thermal stress, partially overlapping with the apoptotic region, with viable cells remaining at the periphery. The outermost region contains cells that were not significantly affected by the burn exposure. 

Figure 8 shows the volumes of injury within skin platforms quantified for different contact burn profiles, with injury categories including the ablated region where the tissue degraded, leaving no cells or matrix behind, the region of dead cells as measured by calcein/PI staining, and the region of apoptotic cells as measured by Annexin V staining. Burns inflicted with a tip temperature of 50 °C were superficial, with volumes so small that they were non-significant from zero. The volumes of the ablated region and dead region were significantly higher (*p* < 0.05) for burns inflicted with a 100 °C tip than for burns inflicted with a 75 °C tip, with the dead region increasing by a factor of 1.8 from 6.9 (75 °C) to 12.4 (100 °C) mm^3^ and the ablated region increasing by a factor of 5.2 from 1.78 (75 °C) to 9.17 (100 °C) mm^3^. The apoptotic region volume was not statistically significantly higher for the 100 °C burns compared to the 75 °C burns, likely due to the large variability observed in results from Annexin V staining for 100 °C burns.

#### 3.2.2. Contact Burn Modeling Results

Modeling the temperature distribution within skin platforms during contact burn testing resulted in the maximum temperature profiles shown in row D of Figure 6, showing the highest temperature reached at each point within the platform. The maximum temperature was used because below a specific ‘breakpoint’ temperature of 43–45 °C, injury from hyperthermia is insignificant and because it has been used as a standard for burn characterization in some past studies [15]. Based on results from our computational modeling simulations, after removal of the heat source (3 s exposure), the heat from the high-temperature regions near the tip dissipated into the surrounding tissue and container, causing a slight increase in the temperature of adjacent tissue, followed by a steady decrease. This is why the simulation was run for a second phase of 1 min beyond the initial 3 s exposure period. CEM43 was calculated from the temperature modeling results and forms contours that roughly match the burned region’s overall shape; however, this metric was found to be a poor indicator of the severity of burn injury. Maximum temperature underpredicted the severity of the high-temperature burns relative to the lower temperatures, whereas CEM43 greatly overpredicted the high-temperature severity relative to lower temperatures. 

The Arrhenius coefficients that best fit the contact burn experimental results for collagen ablation and cell death fit the following relationships: $Ea=2.59\times 10^{3}\;ln\left(A\right)+2.53\times 10^{3}$ for cell death and $Ea=2.58\times 10^{3}\;ln\left(A\right)+8.33\times 10^{3}$ for collagen ablation. The pairs of coefficients that fit these relationships yield close predictions of burn volumes; pairs with higher activation energies have a more linear relationship between predicted volumes and temperature. Within this range, the best fit was found at $Ea=3.26\times 10^{3}$ and $A=2.42\times 10^{54}$ for cell death and $Ea=2.96\times 10^{3}$ and $A=9.65\times 10^{3}$ for collagen ablation, with coefficients of determination of $R^{2}=0.98$ and $R^{2}=0.96$, respectively. Other combinations of process coefficients within this relationship produce similar results. A wide range of values for the Arrhenius coefficients of thermal collagen degradation and cell death were published in previous studies that were used to predict contact burn injury with the computationally modeled time–temperature histories [14,20,59,60]. The coefficients for cell death are relatively similar to those for necrosis reported in some previous studies, and the coefficients for collagen ablation had a similar value for Ea but a substantially lower value for *A*. These coefficients and the coefficients found in the present study using Equation (1) for thermal collagen degradation from low-temperature, long-duration hyperthermia resulted in a significant underprediction of tissue injury, except those found in the present study using high-temperature, short-duration contact burns. This is not unexpected, as many of the coefficients were determined in lower temperature ranges with significantly longer heating durations than those in the present study.

## 4. Discussion

In this study, we present physiologically representative in vitro skin platforms and complementary methods to quantify burn injury severity and temperature distribution for high-temperature, short-duration burn experiments. Multilayer in vitro skin platforms were created from human skin cells embedded within collagen hydrogel scaffolds and then subsequently burned by contact with a cylindrical copper tip at 50–100 °C for 3 s. In previous studies, researchers have used various methods to burn skin models, including conduction through heated metal, scalding from hot liquids, radiofrequency heating, and irradiation with lasers [31,44,47,62,63]. The conduction contact burn injury device used in this study successfully induced consistent skin platform injury, with a range of burn injury profiles spanning from superficial to full-thickness burns. The resulting injury was characterized by quantitatively measuring the volumes of injured tissue with immunofluorescence confocal microscopy, demonstrating consistent injury to skin platforms. 

Many previous skin burn studies did not compare quantitative measurements of burn injury with time–temperature histories and are of minimal usefulness for correlating burn injury with the causative temperatures [4]. Most previous studies quantifying the severity of heat exposure and burn injury used relatively low temperatures in the range of approximately 43–60 °C over more extended periods on the scale of tens of minutes up to 7 h in some cases in order to facilitate the measurement of the time–temperature history [13,20,61,64,65]. Such long-duration burns typically result in reduced injury compared to short-duration burns caused, in part, by the adaptation of cells in response to hyperthermia through the heat shock response [6,10].

Burns caused by high temperatures over short durations, such as the 3 s burns at 50–100 °C investigated in the present study, create temperature distributions within tissue that vary immensely depending on the time and position [13], creating substantial challenges to directly measure the temperature distribution within platforms due to the short distances and timescales involved. We utilized finite element modeling of spatial time–temperature histories within skin platforms during and after contact burn exposure coupled with measurement of the volumes of tissue based on measurements of the spatial distributions of viability and tissue response (complete ablation, cell death, apoptosis, and thermal stress) of burned samples to enable investigation of the mechanisms of burn injury. This enables quantitatively determination of the relationship between thermal injury and temperature history for high-temperature, short-duration burns, facilitating predictive models for skin burns. Thermal injury rates and thresholds can then be studied at the temperatures of interest without the need to extrapolate from lower-temperature experiments or the uncertainties inherent in such an approach [10].

We comprehensively characterized the developed skin platforms to validate their practicality as in vitro models of skin, evaluate changes throughout experiments, and determine their thermal properties for use in in silico modeling. Thermal conductivity and specific heat capacity measurements from the in vitro skin platforms were used as inputs to a finite element model to spatially predict temperature distribution in the platforms during contact burn experiments, allowing the time–temperature history to be quantified. Tissue injury was defined by temperature and time, not the total heat energy used. Therefore, the correlation of experimentally observed injury and the computationally modeled temperature is valid if the thermal characteristics are known for the platforms [4,5]. Tissue stiffness, as represented by the compression modulus, influences cell growth and affects how high-temperature, short-duration burns injure tissue due to the thermally induced deformation of tissue [10,12]. The average compression moduli of multilayer skin samples recorded in the present study were found to be 20.01 kPa. This value is very close to the 24.910 kPa recorded by Liang et al. for human palm skin [66] and within the broader average value range of 7 to 344 kPa published in the literature for human skin in various regions of the body, as measured in the perpendicular direction [54,67,68,69]. The thermal conductivity of our skin platforms was found to be 0.603 ± 0.128 W/mK. This value is close to the thermal conductivity of water at 25 °C (0.606 W/mk) but is higher than the measured thermal conductivity of human skin reported in previous studies (0.293–0.322 W/mK) [70,71]. This discrepancy is accounted for by the computational model of temperature throughout the samples and does not affect the resulting Arrhenius analysis.

Skin platforms were stable for three-week experiments, and compression moduli and thermal conductivities of skin platforms exhibited no measurable changes during this period. This consistency negated such changes as a possible source of error between trials and over longer-term experiments. While the burn experiments in the present study were conducted 24 h after platform creation, we expect similar results would be observed on 3-week-old platforms, given that the characteristics governing the flow of heat did not significantly change.

Results from long-duration, low-temperature hyperthermia yielded Arrhenius process coefficients of A= 1.61 × 10^45^ s^−1^ and Ea=2.96 × 10^5^ J/mole for thermal collagen degradation, both of which are within 3% of the values published by Pearce et al. for thermal degradation of rat cartilage collagen [13]. This supports the method of measuring reductions in compression modulus as an indicator of degradation and suggests a strong similarity between the matrix material of the in vitro skin tissue platforms reported in the present study with those reported in previous works. Additionally, long-term hyperthermia results suggest that CEM43 is a poor predictor of collagen matrix degradation for higher temperature ranges. This shortcoming has been demonstrated in previous studies and is consistent with the prevalent trends observed with Arrhenius models [6,69]. There are many different burn injury mechanisms, each defined by its own set of processes, and the relationship between temperature and rate of injury differs between them. To develop practical predictive models of burn injury, distinct isoeffective metrics must be used for each type of injury, each with its own rate coefficients and breakpoint temperatures [13]. The skin platforms and methodology presented in this paper determine the time–temperature history, which is related to many different types of thermal injury, enabling new predictive models to be developed.

The combined approach of experimental contact burns with analysis enlightened by computational modeling to determine temperature was used in a previous study by Orgill et al. [71]. In vivo pig skin was burned by conduction; then, the resulting injury was analyzed by biopsy and staining to determine the depth of injury. The injury was predicted using the Arrhenius model with process coefficients from another previous study by Rylander et al. [8], with finite element modeling using published thermal properties of skin. The results reported by Orgill et al. support the validity of the combined experimental and computational approach, although they failed to account for several factors that limit the applicability of their data for predictive models. In vivo pig skin is highly variable and differs from human physiology, and the lack of measurement of the thermal characteristics of their samples introduced error. Additionally, measuring depth only neglects the three-dimensional nature of burn injuries. We addressed these shortcomings in the present study by using consistent in vitro skin platforms with human cells and measuring thermal properties, as well as by using volume measurements that account for the entire burn injury rather than just the deepest point. In this way, consistent quantitative results were obtained, which are invaluable for the development of predictive models. 

As expected, applying the Arrhenius process coefficients (A and $E_{a}$) determined in this study from low-temperature, long-duration hyperthermia to high-temperature, short-duration injury resulted in underpredictions of burn injury. The finite element model is unlikely to have underdetermined the temperature within platforms because it does not account for the heat absorbed by endothermic reactions; thus, this disparity is indicative of the lack of applicability of low-temperature results to high-temperature conditions. Applying the Arrhenius process coefficients from numerous previous studies to predict contact burn injury also resulted in underpredictions, as the coefficients were determined using different temperature ranges and injury mechanisms [14,20,59,60,72]. Determining Arrhenius process coefficients from high-temperature, short-duration contact burns resulted in activation energy ($E_{a}$) values that were substantially lower than those obtained from low-temperature, long-duration hyperthermia, consistent with findings from previous studies [73]. As measured by calcein and propidium iodide staining, the activation energy parameter determined for thermal cell death is the closest match to results from previous studies of the evaluated metrics. The relationship between $E_{a}$ and A in the present study results in coefficients of determination of 0.96 and 0.98 between experimental and predicted burn volumes. However, the frequency factor (A) is substantially greater than that suggested in those works. Overall, these results support the conclusion that as temperatures increase, the mechanisms of burn injury change and that the extrapolation used in many previous studies according to results obtained at lower temperatures only to higher temperature burns is problematic [8,73]. Experiments must be performed in the temperature ranges of interest to obtain the best results.

The Arrhenius model derived in the present study can be used to predict the severity and distribution of burn injury on skin models with different structures and geometries if the flow of heat through the samples can be determined. Time–temperature history is the most important factor for burn injury and is the only input to the Arrhenius model with known process coefficients [3,4,5]. The geometry of the resulting burns varies depending on the type of skin model, but at every point, the time–temperature history yields consistent results. This likewise applies to other modalities of heating, such as radiation or scalding, to account for varying kinetics of the thermal source, which require finite element models of temperature but otherwise yield similar results. While cells themselves may have significantly different thermal properties than their surroundings, they comprise an insignificant volume of the skin platforms and thus have minimal effect on the flow of heat [73,74]. Other studies that use in vitro engineered skin platforms for various applications may culture the keratinocytes in the epidermis layer with an air–liquid interface, stimulating the differentiation of the epidermis into a stratified structure after approximately 17 days [75]. While these differentiated epidermis layers are more representative of in vivo human skin, they do not provide more utility in the collection of the quantitative time–temperature and burn response data needed to develop Arrhenius models. By using undifferentiated epidermis layers in the present study, we obtained the required data using samples that can be prepared in 24 h rather than several weeks. 

The results of this study demonstrate how multilayer skin platforms can be utilized to quantitatively evaluate how human skin is affected by high-temperature contact burns. The spatial and temporal temperature data with corresponding tissue response enables the process of contact burn injury to be modeled using the Arrhenius equation, which is invaluable for the development of predictive models for improved understanding and treatment of burns. Unlike previous studies, the presented work accounts for the three-dimensional propagation of heat and the resulting range of injury severity from complete tissue ablation at the center of the contact burn expanding outward in layers through dead, apoptotic, heat-stressed—as evidenced by HSP expression—and ultimately unaffected cells at the periphery. No previous studies have determined the spatial distributions of temperature and the resulting tissue injury for short-duration, high-temperature burns and used these data to calculate the governing Arrhenius coefficients for the range of injury severities. The use of stained tissue sections allows for limited insight into these processes, but significantly less information is obtained per sample, which is susceptible to greater variability due to the selection of individual 2D slices. Some studies have measured the spatial distribution of temperature and HSP expression within burn models but did not calculate Arrhenius coefficients except for low-temperature, long-duration hyperthermia experiments with mostly homogenous distributions [61,64]. These distinct injury types are each governed by their own set of process coefficients. Comparison of the volumes of injured tissue at different burn temperatures provides information valuable for understanding the underlying burn injury mechanisms and their temperature dependencies. Platforms can also be adapted for a wide range of testing applications, including different burn modes, tissue dimensions and morphologies, cell phenotypes, disease states, and even other tissue types entirely. Future and ongoing work for our group involves the inclusion of parallel vascular channels in the dermis layer with media circulation to investigate the influence of blood perfusion on skin burns. Additional research is needed to develop new metrics that take into consideration the difference in mechanisms between types of burn injury. The rapid growth times and versatility of the in vitro skin tissue platforms and experimental methods presented in this study make them ideal for the large volume of research required for this purpose. 

## Figures and Tables

**Figure 1 bioengineering-10-00265-f001:**
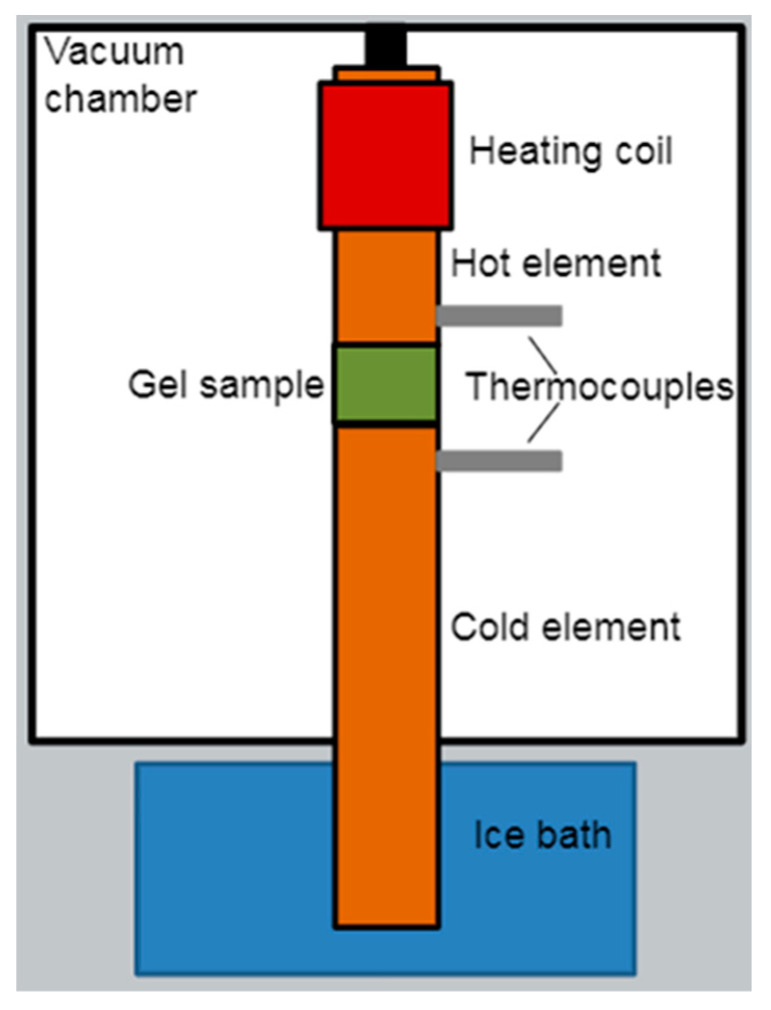
Diagram of the apparatus used to measure the thermal conductivity of hydrogel samples. A constant heat input of 0.66 W is applied to the hot element using the heating coil. This heat travels through the hot element, the gel sample, and the cold element and into the ice bath. When the system reaches a steady state, the hydrogel’s thermal conductivity is calculated using the temperature difference on either side of the sample, the known power output, and the sample’s surface area and thickness.

**Figure 2 bioengineering-10-00265-f002:**
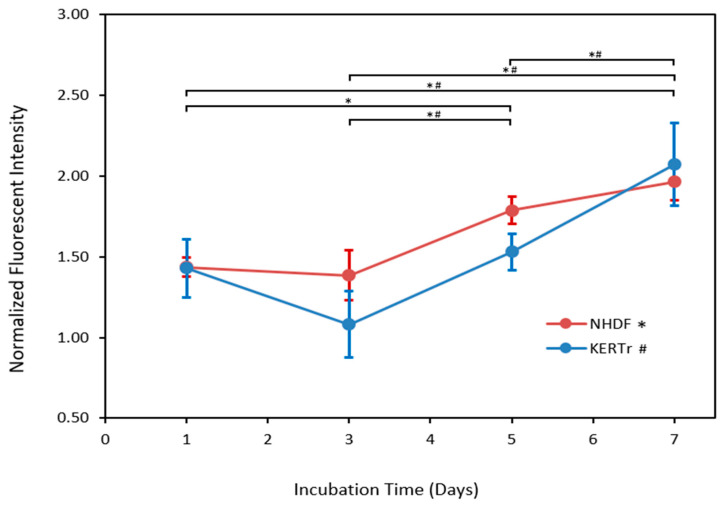
Normalized fluorescent intensity of single-layer in vitro skin platforms over time, as measured by CellTiter Blue assay. Data are normalized relative to results from day 0. Data are shown as average values ± standard deviation (*n* = 3); * and # indicate statistical significance at *p* < 0.05 for NHDF and KERTr, respectively.

**Figure 3 bioengineering-10-00265-f003:**
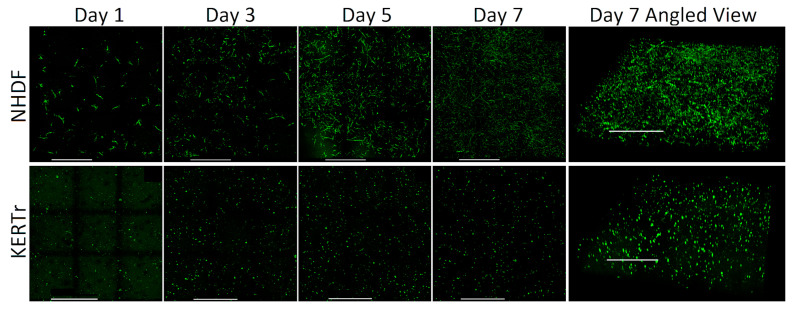
Spatial distribution of NHDF and KERTr cells within single-layer skin platforms imaged with immunofluorescence confocal microscopy. The first four columns from the left show the top view of cells within the platforms over time. The last column shows an angle view of platforms on day seven. Scale bar = 1 mm.

**Figure 4 bioengineering-10-00265-f004:**
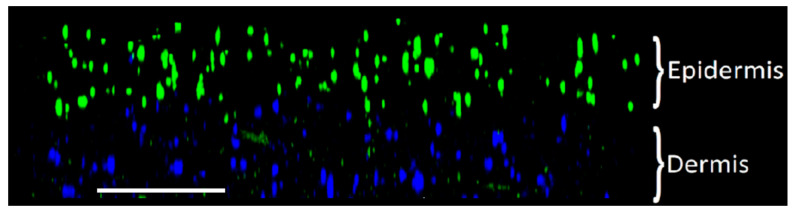
Side view of a multilayer in vitro skin platform seven days post sample preparation imaged with immunofluorescence confocal microscopy. KERTr and NHDF are colored green and blue, respectively. The distinct dermal and epidermal layers are evident in their appropriate layers. Scale bar = 1 mm.

**Figure 5 bioengineering-10-00265-f005:**
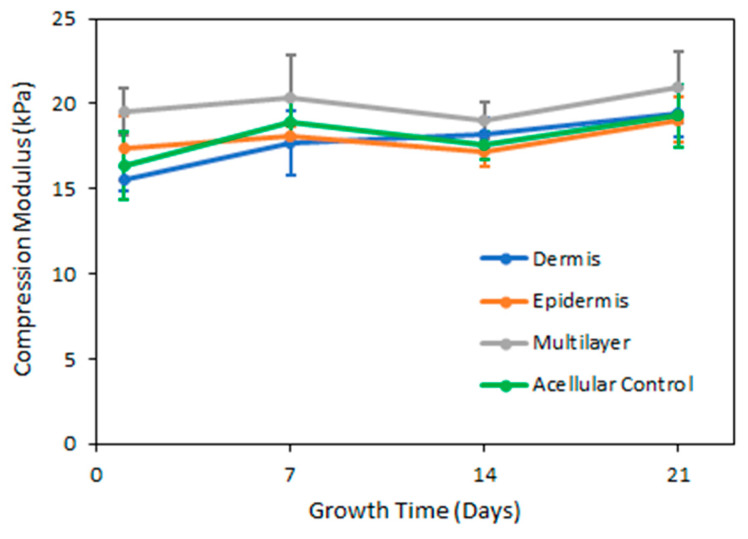
Compression modulus of in vitro skin platforms evaluated after 1, 7, 14, and 21 days of cell growth. Epidermis and dermis platforms are single-layer platforms containing KERTr or NHDF, respectively. Cell-bearing multilayer platforms have an epidermis layer with keratinocytes atop a dermis layer with fibroblasts, with the dermis layer three times thicker than the epidermis. Data are shown as average values ± standard deviation (*n* = 4). None of the platforms were significantly different from the control.

**Figure 6 bioengineering-10-00265-f006:**
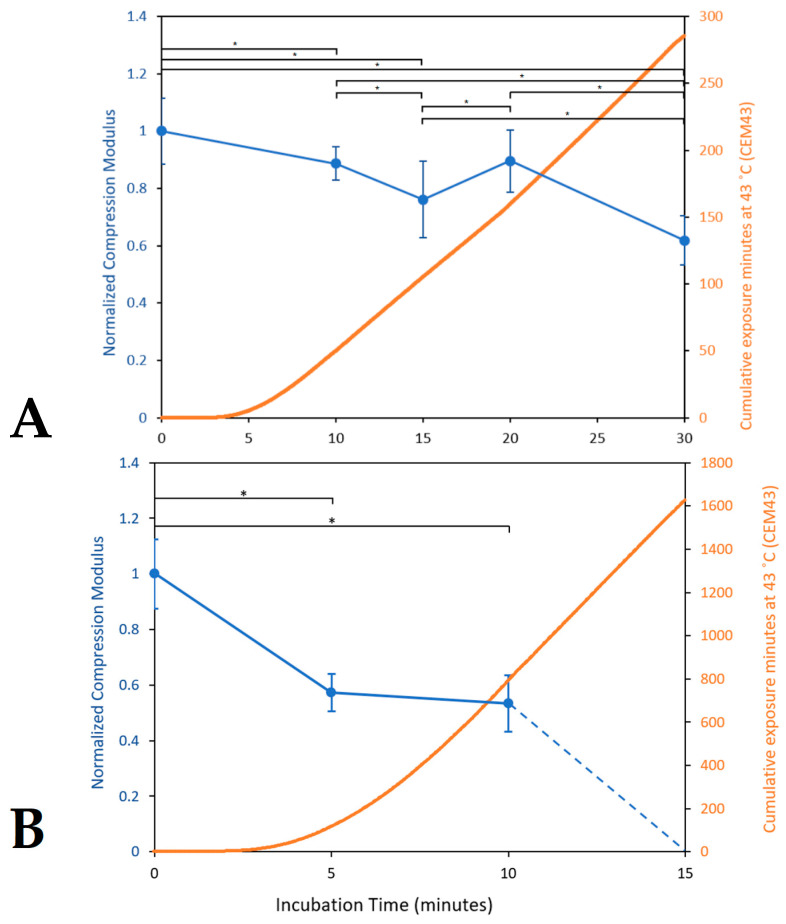
Normalized compression moduli and thermal dose quantified by cumulative effective minutes at 43 °C (CEM43) of acellular platforms after incubation at hyperthermic temperatures for different lengths of time. (**A**): 47 °C exposure (*n* = 4); (**B**): 50 °C exposure (*n* = 4). Data are shown as average values ± standard deviation; * indicates statistical significance at *p* < 0.05.

**Figure 7 bioengineering-10-00265-f007:**
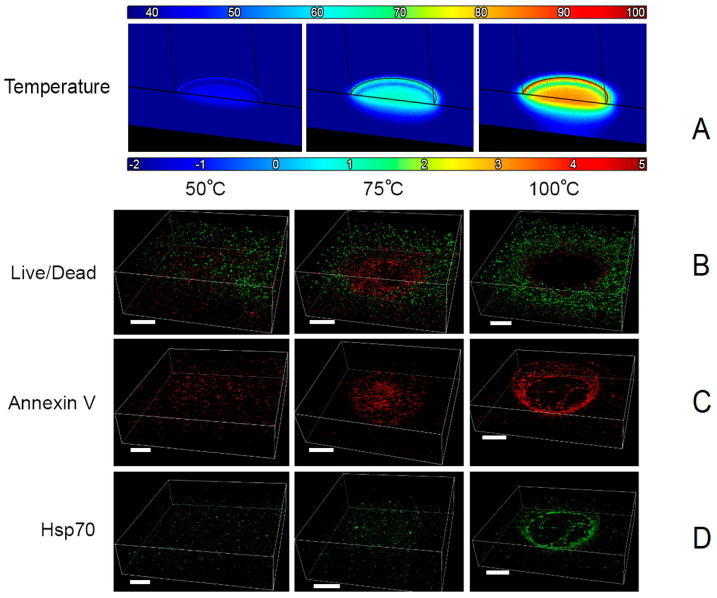
Three-dimensional images of multilayer skin platforms 24 h after exposure to contact burns (scale bars = 1 mm). Row (**A**): computational prediction of maximum temperature reached within platforms during and up to one minute after exposure; the outline shows the position of the burn tip during exposure. Row (**B**): viability stained with calcein/propidium iodide; live and dead cells are green and red, respectively. Row (**C**): apoptotic cells stained with conjugated Annexin V. Row (**D**): Hsp70 expression.

**Figure 8 bioengineering-10-00265-f008:**
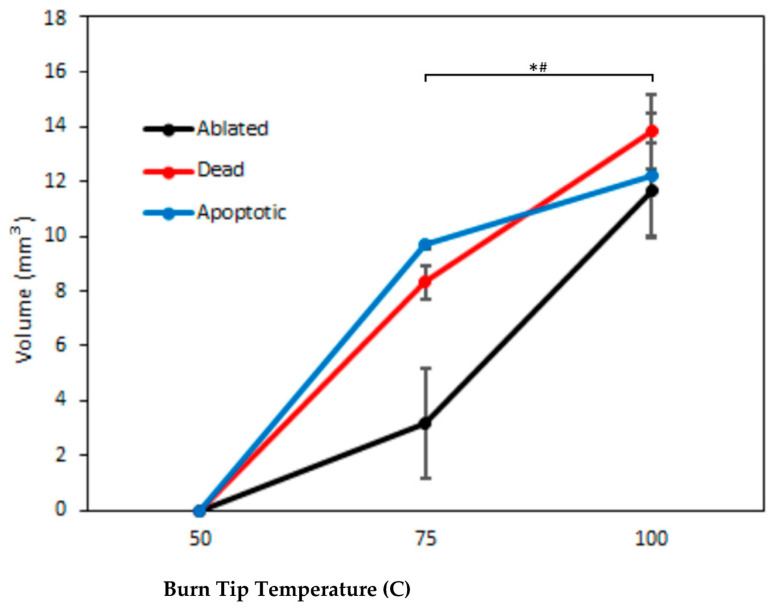
Volume of injury to skin platforms caused by 3 s of exposure to various contact burn tip temperatures, defined as follows: ablated region, where the matrix is no longer present; dead cells, measured by calcein/PI staining; apoptotic cells, measured by Annexin V staining. Data are shown as average values ± standard deviation (*n* = 3); * and # indicate significance at *p* < 0.05 for the ablated and dead regions, respectively.

## Data Availability

The data presented in this study are available on request from the corresponding author. The data are not publicly available due to the large size and quantity of data files.

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
