# Peer review of "Multilayer In Vitro Human Skin Tissue Platforms for Quantitative Burn Injury Investigation"

_bioengineering, 2023, doi:10.3390/bioengineering10020265_

Round 1

Reviewer 1 Report

The manuscript by Brocklehurst et al. deals with the interesting topic of tissue/cell stress response to extreme conditions, specifically contact burns. The authors used the in vitro skin model to eliminate the inherent variability of ex vivo skin samples and the low relevance of simple cell monolayers. The methods used to evaluate this model include both molecular biology and mechanic measurement.

From my point of view, there are some outstanding questions that should be answered first. Without addressing these potential problems the paper is not suitable for acceptance in this journal. In case those are solved in the revised version of the manuscript, the topic is of interest and the acceptance can be considered.

1.         Why hasn’t the classic reconstituted skin been used? the use of collagen 1 hydrogel to create the “epidermal” layer is questionable. The morphology, physiology, and theoretically also the response to the burn can be affected by the stratified and cornified features of top epidermal layers. Using the gel-embedded keratinocytes in submerged conditions you cannot achieve this epidermal feature in your testing system. This should be included in the discussion part.

2.         Authors stated that the collagen-based skin model is stable for 3 weeks based on its viability and mechanical properties. But the main functional parameter is the model's response to the heat injury. Thus, the data from “fresh” (app 24 hours old) and “aged” (app 3 weeks old skin models should be measured and reported. There is no information if senescence and other potential changes occurring during those 3 weeks of cultivation that will affect cell response to the injury.

 These points should be addressed first as they are parts of the whole experimental design and affects the interpretation of any obtained results. The rest of the results, the way they were obtained and discussed are acceptable.

Author Response

detailed responses are appended to the manuscript file

Reviewer 2 Report

This is a well designed experimental research; the manuscript while somewhat lengthy is well written.

However , the authors have not dicussed possible shortcommings of their model. For the reader who is not familiar with burns some restrictions should be mentioned. These should be discussed in a separate section:

different thickness of skin esp dermis

scald vs contact vs flame burn

3 D two cell component model vs human skin WITH blood flow 

cell appendages missing in 3D model affecting thermal transmission most important for healing

Author Response

(The authors gave the same response as above.)

Round 2

Reviewer 1 Report

Changes to the manuscript and explanations provided by the authors are sufficient. As my main concerns has been addressed I can recommend this manuscript to be accepted for publication in Bioengineering journal.

Congratulations.